# Overcoming Hurdles in Nanoparticle Clinical Translation: The Influence of Experimental Design and Surface Modification

**DOI:** 10.3390/ijms20236056

**Published:** 2019-11-30

**Authors:** Jacob W. Shreffler, Jessica E. Pullan, Kaitlin M. Dailey, Sanku Mallik, Amanda E. Brooks

**Affiliations:** Department of Pharmaceutical Sciences, North Dakota State University, Fargo, ND 58105, USA; jacob.shreffler@ndsu.edu (J.W.S.); jessica.pullan@ndsu.edu (J.E.P.); kaitlin.dailey@ndsu.edu (K.M.D.); sanku.mallik@ndsu.edu (S.M.)

**Keywords:** nanoparticle clearance, surface modifications, protein corona, animal model selection

## Abstract

Nanoparticles are becoming an increasingly popular tool for biomedical imaging and drug delivery. While the prevalence of nanoparticle drug-delivery systems reported in the literature increases yearly, relatively little translation from the bench to the bedside has occurred. It is crucial for the scientific community to recognize this shortcoming and re-evaluate standard practices in the field, to increase clinical translatability. Currently, nanoparticle drug-delivery systems are designed to increase circulation, target disease states, enhance retention in diseased tissues, and provide targeted payload release. To manage these demands, the surface of the particle is often modified with a variety of chemical and biological moieties, including PEG, tumor targeting peptides, and environmentally responsive linkers. Regardless of the surface modifications, the nano–bio interface, which is mediated by opsonization and the protein corona, often remains problematic. While fabrication and assessment techniques for nanoparticles have seen continued advances, a thorough evaluation of the particle’s interaction with the immune system has lagged behind, seemingly taking a backseat to particle characterization. This review explores current limitations in the evaluation of surface-modified nanoparticle biocompatibility and in vivo model selection, suggesting a promising standardized pathway to clinical translation.

## 1. Introduction

Although millions in research funding has been allocated to nanoparticle drug development, the translation from research laboratory to clinical implementation, particularly for drug delivery nanoparticles, remains limited. This level of funding support is reflected in the publication statistics. There were over 20,000 reports published in 2018 involving nanoparticles. Of these, only 4700 reported nanoparticle-involved drug delivery (Figure 1A), with even fewer progressing to clinical translation. Of the current clinical trials being performed for surface modified nanoparticles used in drug delivery, many are failing to complete phase II for a variety of reasons (Figure 1B). Notably, there are other successful applications of nanoparticles, particularly for imaging and diagnostics. Although the reasons for the somewhat measured progress of nanoparticles in drug delivery are multifaceted, they are significantly driven by (1) the nanoparticle surface and resulting protein corona, which can alter drug release, (2) high immune clearance rates that compromise targeting, and (3) a lack of translatability both in vitro and in vivo, as well as between animal models and clinical patients. While the nanoparticle field in general, and laboratory-scale surface modified nanoparticles specifically have led to promising results with numerous advantages for drug delivery, ultimately, surface modified nanoparticles have fallen short of their promise to mitigate the rapid clearance and pathological interactions of the particle with the biological system (Figure 2).

Many pathological reactions to nanoparticles are a direct result of clearance by the immune system, which limits retention time in the circulatory system. Far too frequently, in vitro and in vivo studies fail to properly account for the immune system’s effect on nanoparticle progression toward targeted drug delivery, contributing to low clinical translation rates. Furthermore, while animal models are a necessary step in clinical development, they have yielded contradictory results based on the species and may offer limited value when converting into human application. The significant hurdles outlined above, in addition to the skyrocketing costs of regulatory approval, can deter many scientists from pursing nanoparticle therapeutic development beyond proof-of-concept small animal models. This review explores the limitations that have plagued the clinical translation of nanoparticles, specifically discussing the influence of the immune system and preclinical model selection on nanoparticle drug delivery (Figure 3). Finally, several recommendations for preclinical model selection are posited to improve the clinical translation of drug-delivery nanoparticles.

## 2. Surface Modifications

In order to combat traditionally high nanoparticle clearance rates, a diverse array of bulk formulations—including natural materials (i.e., chitosan [1], dextran [2], liposomes [3], protein [4], and exosomes [5]), carbon based carriers [6,7], branched polymers [8], polymer carriers (i.e., polymersomes [9,10], block copolymers [11,12], and dendrimers [13,14]), ferrofluids [13,14], quantum dots [4], and inorganic nanoparticles [15,16,17]—have been tested for many types of biomedical applications, including drug delivery and imaging [18,19]. However, despite the inherent and unique advantages of each of these materials, all unmodified nanoparticles suffer from high and nonspecific clearance. Hence, surface modifications and functionalizations applied to the bulk nanoparticle material are designed to either mask intrinsic characteristics of the nanoparticle or manipulate biological interactions with the particle. These modifications can be divided into modifications that (1) increase residence and circulation time, (2) target a desired tissue, and (3) selectively deliver a pharmaceutical payload to a target tissue.

### 2.1. Modifications to Enhance Circulation

Rapid and nonspecific clearance is a ubiquitous problem for nanoparticle-based drug delivery. To increase circulation time, bulk nanoparticle materials can be chemically modified with a variety of compounds, including polyethylene glycol (PEG) and its derivatives (e.g., methoxy PEG (mPEG)) and acetyl groups [20,21]. PEGylation, perhaps the most common chemical modification, is the addition of polyethylene glycol to the surface of a nanoparticle generally based on the hydrophobicity of the material [22]. Acetylation is an alternative surface modification with similar retention affects. In a study examining the difference between PEGylation and acetylation [23], both modifications demonstrated increased retention time in nontargeted PAMAM dendrimers [23]. In fact, retention times did not differ significantly; however, the targeting ability of the particle was negatively affected by acetylation in an in vitro tumor cell model [23]. There are hundreds of examples, some in clinical trials and some that are already Food and Drug Administration (FDA) approved (examples provided in Table 1), of nanoparticles utilizing surface attached PEG.

Factors such as PEG molecular weight and amount of incorporation can drastically affect nano–bio interface by increasing the nanoparticle size or surface charge density [20,36,37]. Regardless, decorating the surface of a nanoparticle with PEG is hypothesized to increase tumor accumulation, circulation time, and specific cellular uptake by allowing the nanoparticle to move through the bloodstream to its destination with minimal interference by the immune system [16]. For example, PEGylated gold nanoparticles have shown an increased blood half-life with the addition of PEG in direct relation to the molecular weight of the attached PEG. Specifically, PEG_5000_ increased the blood half-life up to 16.5 h while PEG_2000_ only showed a modest improvement to 2.5 h. Additionally, PEGylation has been shown to decrease both nanoparticle aggregation (a relatively consistent struggle for nanoparticle stability) and protein opsonization (i.e., proteins will bind to the nanoparticle, tagging it for immune system clearance) [16]. The decrease in protein opsonization corresponds to an increase in the nanoparticle’s circulation time [21]. The increase in circulation time is thought to not only be a byproduct of the particle’s increased stability and decreased opsonization, but is a result of the overall masking or “stealthing” of the particles. Comparative studies have been performed between PEGylated and un-PEGylated versions of the FDA approved drug Onivyde [21]. Although the PEGylated version of Onivyde performed better, the advantages seemed to be based more on the liposomal surface chemistry than its ability to be “stealthy” while in the blood stream [21]. This is not the first study to explore the benefit of PEGylation as a modification method to increase retention time. Conversely, several studies have been performed to understand significant reductions in effective drug delivery from PEGylated nanoparticles, finding enhanced serum protein binding, reduced cellular uptake, and an elicited immune response [38]. In spite of these increasingly prevalent hints that PEGylation may not be a panacea for drug delivery and the fact that additional statistically powered studies remain to be done, PEGylation remains a popular, and often effective, initial modification to improve nanoparticle delivery [38].

Not only has the efficacy of PEGylation recently come under fire, but the safety of repeated exposure to PEG is also being called into question. Repeated exposure to PEG compounds is of significant concern since the use of PEG in drugs, cosmetics, etc. has climbed almost exponentially since its discovery. These risks are highlighted by the rise of antibodies against PEG (0.2% to 0.25%) found in both common animal model systems and in humans over the last twenty years [39,40,41,42]. Importantly, while PEG antibodies are now routinely found in humans (although the data remains somewhat controversial potentially due to a lack of reporting [43,44]), their presence is often neglected in preclinical animal models [40,41,45,46,47]. Notably, increased detection capabilities and prevalence of PEG containing products cannot be excluded as significant contributing factors in this drastic rise. Regardless, it is clear that PEG antibody concentrations are increasing and must be considered in the development of modern pharmaceutics, as shown by Grenier et al. (Figure 4) [48]. Grenier’s group not only saw increasing concentrations of anti-PEG antibodies mostly from IgM but also noted increased clearance rates of PEGylated nanoparticles over the course of 11 days (Figure 4) [48]. Additional studies have shown that, in the presence of anti-PEG antibodies, PEGylated nanoparticle clearance rates increase drastically [44,45,49,50]. Increased clearance is likely due to a pathological protein corona that mediates the immune response and shuttles the nanoparticle down an alternative pathway [51,52], potentially compromising the safety of this approach. Importantly, although the biological response to PEG may change based on the molecular weight and surface density of PEG, once PEG antibodies are present in the system, any modifications made to the PEG moiety (or linkers attached) are irrelevant [53]. This renders the modifications not only inconsequential to stopping the rapid clearance of the particle, but can also negatively affect in vivo testing if not properly negotiated [53].

Aside from the nanoparticle’s bulk composition, surface modifications beyond PEGylation have been shown to significantly diminish obstacles associated with targeting and retention, specifically the challenges associated with MPS (mononuclear phagocyte system) clearance. Although PEGylation is perhaps the most fundamental nanoparticle modification, new controversy surrounding its efficacy and safety in conjunction with insights into cellular microenvironments have led to modifications of the nanoparticle surface with proteins. Modification of the nanoparticle surface with proteins such as small targeting peptides (e.g., RGD) and ubiquitous blood component (e.g., albumin) effectively masks the synthetic, foreign surface of the particle with a biological mimetic [39,40,41,42]. Based on its prevalence in the plasma, albumin is one of the most commonly chosen proteins. Albumin offers a host of other advantages, as well. Not only is it easy to obtain, easy to use, and low in cost, but albumin also has a proven track record of both regulatory approval and beneficial effects on drug loading and release [54]. Furthermore, albumin often has the added benefit of stabilizing the drug by increasing its half-life. FDA-approved albumin-bound paclitaxel nanoparticles (Abraxane) are paving the way for new protein-modified nanoparticles. Since Abraxane’s approval, the field has exploded with options to exploit protein-coated nanoparticles in order to further increase targeting and retention. In general, protein-based nanoparticles have several desirable characteristics, including but not limited to biodegradability, lack of immunogenicity, lack of toxicity, improved drug solubility, enhanced circulation time, preferential uptake in tumor and inflammatory tissues, and a stable structure across a range of pH and/or temperatures [54]. Other proteins that are being explored for nanoparticle drug delivery include heat-shock proteins [55], silk proteins [56], soy proteins [57], collagen [58], elastin [59], gelatin [60], and VEGF [61].

### 2.2. Modifications for Cellular Targeting and Retention

In addition to using structural and biologically active proteins to enhance circulation, protein modifications can also improve the retention and targeting of drug-carrying nanoparticles. The use of tumor-targeting and cell-penetrating peptides to modify a nanoparticle surface has experienced a renewed surge of popularity [62,63], with the most widespread peptides being RGD, iRGD, and iNGR [64,65,66]. These peptides rely on the upregulation of specific ligand receptors (e.g., neuropilin-1) commonly found in solid tumors [65]. The use of tumor-targeting and cell-penetrating peptides for liposome and polymersome drug delivery, particularly for solid tumor cancers, is becoming an increasingly common strategy [9,66]. In addition to neuropilin-1 binding peptides, Epidermal Growth Factor Receptor binding peptides, integrin binding peptides, Vascular Endothelial Growth Factor binding peptides, guanine nucleotide exchange factor binding peptides, protein tyrosine phosphatase receptor type J binding peptides, platelet derived growth factor receptor binding peptides, and interleukin receptor binding peptides have all been targeted [62]. While most of these peptides target proteins upregulated on tumorigenic cells, intense research continues to find other disease specific targets. Despite the collective efforts of the scientific community, off-target protein binding is still considered the biggest, and perhaps most critical, pitfall of these surface modifications.

To improve the odds of specific binding, thereby limiting off target effects, tethering antibodies to the nanoparticle surface is a promising alternative [67]. Despite the theoretical ease with which the strategy can be employed, it is technically challenging to produce an active surface-bound antibody due to the intrinsic properties of antibodies (e.g., epitope display, etc.) that must be protected during modification. Perhaps one of the most elegant examples of the strategy can be found in the recent study of Su et al. The investigative team targeted the HER2 receptor, known to be overexpressed in breast cancer cells, using PEGylated micelles encapsulating docetaxel. The micelles were further modified by tethering an anti-HER2 antibody to the surface of the micelle via an anti-mPEG linker [68]. This strategy yielded promising results in MCF7 breast cancer cells and demonstrated the power of using antibody targeting as a tool. Unlike many other types of modifications, where the available data are biased to cancer applications, the efficacy of this approach applies to a wide range of diseases. For example, other recent studies have shown impressive results using siRNA containing chitosan nanoparticles targeted to HIV infected astrocytes [69]. Despite dramatically improving nanoparticle targeting, the large size of antibodies and other proteins increased the overall nanoparticle size (average diameter of 235 nm), which will enhance nanoparticle clearance by the MPS [69].

The use of biologically derived molecules provides clear advantages for targeting and in some cases retention; however, the use of synthetic dendrimers for drug delivery cannot be ignored, as they offer some distinct advantages in the context of retention. Not only does the characteristic treelike structure of a dendrimer generally increase retention time, but the presence of a large number of surface functional groups also allows the particle to be effectively modified for targeting. Furthermore, dendrimers have excellent water solubility, as well as areas of increased hydrophobicity, allowing them to be very effective at carrying a drug payload [70]. Despite these advantages, dendrimer drug delivery is not a magic bullet, as it has a long production time and contradictory results [71,72,73]. While the unique chemical synthesis of dendrimers allows for more autonomy in design, few dendrimer designs vary from standards [70]. The success and limitations of these standard designs may explain why the use of this particular class of nanoparticles for drug delivery is limited in favor of more diagnostic applications [70].

Cell-specific targeting has been a revolution in drug delivery, drastically reducing the side effects of powerful pharmaceutical compounds. Nevertheless, both the selection of the target as well as the mechanism of targeting must be deliberate, with careful consideration given to systems biology and the cellular microenvironment. Although the search for the perfect target continues within the field, the advanced ability to camouflage a synthetic nanoparticle surface with a biological cloak to both increase retention time and utilize cellular trafficking to target drug delivery represents a substantial step in the right direction.

### 2.3. Modifications Targeting Payload Release

Beyond general tissue targeting, the safety of nanoparticle drug-delivery systems can also be improved by integrating a targeted, microenvironment responsive release strategy. Strategies utilizing tissue- and disease-specific microenvironments to trigger drug release have gained widespread attention in the treatment of various diseases (e.g., cancer, diabetes, and bacterial infections). Microenvironment activated drug release increases drug stability and therapeutic efficacy while decreasing toxic side effects [74,75,76,77]. Particularly useful microenvironmental biosensors include physical factors such as pH, ion sensitivity, hypoxia, and enzyme abundance. The pH and ion sensitivity represent more ubiquitous and less-restrictive triggers, often being significantly influenced by the chosen route of administration (ROA) [10]. For example, the ROA demonstrated an effect on payload release in a study of an anionic polymer created to be responsive to high intestinal pH, in order to protect the treatment-drug cargo from gastric degradation [10]. Importantly, the required ROA can also impact the choice of preclinical model used for in vivo evaluation. Modifications to sensitize nanoparticle-payload release to the biology’s pH and ionic concentration are valuable methods; however, the conceptual design principles used expose critical flaws in the strategy that must be considered. Primarily, variation in pH and ionic concentration occur in multiple locations throughout the body, such as in the GI tract, which can lead to injurious release outside the target location (i.e., off-target effects). However, a substantial body of work indicates that, when nanoparticles do reach their destination, they can successfully release a drug payload [60]; hence, coupling pH and ionic sensitive release with a targeting moiety may be a necessary modification to improve the success of the strategy. PEGylated ZnO quantum dots carrying doxorubicin provide a powerful example of a pH- and ion-sensitive drug-delivery system [78]. The ZnO quantum dots degrade to produce a Zn^2+^ ion when placed in an acidic environment similar to that of a lysosome or endosome, thereby triggering the release of doxorubicin upon cellular uptake [78]. Although quantum dots are not traditionally used for drug delivery, their ability to successfully deliver doxorubicin to lung cancer cells by using this strategy warrants further consideration and their inclusion within this review. In another compelling example of a nontraditional particle being used for drug delivery, a combination of generation-two dendrimers and magnetic nanoparticles was used to deliver methotrexate [79]. In this sophisticated implementation of pH-triggering technology, the authors loaded the combination nanoparticles with high-concentration methotrexate and demonstrated pH-controlled drug release in MCF-7, HeLa, and Caov-4 cell lines [79]. However, it must be noted that promising in vitro results often fail to translate into in vivo success, so more work remains to be done. A similar pH-triggered drug-release strategy was explored by using a more traditional hollow mesoporous silica nanoparticle (HMSN) [80]. In this particular study, HMSN were modified with pH-sensitive bornoic acid–catechol ester bonds (pH 5.0) to encapsulate doxorubicin within the mesoporous structure. Subsequently, the particles were PEGylated with a pH-sensitive benzoic-imine bond (pH 6.8), to protect them. As predicted, the particles were able to release doxorubicin, showing promising results both in vitro and in vivo in a nude mouse model.

While pH and ion sensitivity are perhaps the more traditional triggers for drug release, hypoxia and enzyme concentration are usually considered more specific signals and may be perceived more favorably. Hypoxia-responsive nanoparticles are becoming an increasingly familiar strategy because of the hypoxic microenvironment consistently found within solid tumors, regardless of tissue origination. The use of a reducing lipid to modify the nanoparticle sensitizes the nanoparticles to excess electrons found within the hypoxic, acidic environment, breaking the chemical bond (frequently a disulfide or azo bond) and allowing a burst release of the therapeutic payload [66,81,82,83,84].

Although hypoxia-sensitive particles are more specific than pH- or ion-responsive particles, modifications to render drug-delivering nanoparticles susceptible to an enzyme are perhaps the most precise type of targeted payload delivery. Enzyme-triggered drug release is one of the most promising approaches for biologically responsive drug delivery [28,39]; however, its efficacy is inextricably connected to (1) identification of a disease specific protease, (2) identification of a peptide sequence with high affinity for the protease, (3) specificity of the protease recognition site, and (4) compatibility of the polymer matrix carrier. Many disease conditions (e.g., cancer [85], cardiovascular injury, inflammation [86], and bacterial infection [75,87,88]) are characterized by abnormal and specific protease upregulation [89,90]. Infection has been associated with upregulation of phosphatase or phospholipase [75], esterase [91], or thrombin-like enzymatic cleavage [92]. However, just as in other mechanisms of payload release, cancer is the most frequently researched disease state. One study used mesoporous silica nanoparticles encapsulating doxorubicin with triphenylphosphine (TPP) and hyaluronic acid (HA) grafted to the surface [93], not only demonstrating uptake by cancer cells via CD44 receptor-medicated endocytosis but also release of doxorubicin in response to HAase cleavage, ultimately decreasing cell viability to 34% [93]. This study demonstrates how effective enzyme responsive payload release can be. Alternatively, several different microenvironment triggered-release strategies have been explored for infection, particularly pathogenic *S. aureus* infection. Recently, Yang et al. reported a sophisticated strategy to combat intracellular *S. aureus* by using a dual modified lipid bilayer coating an underlying mesoporous silica nanoparticle. The nanoparticle was modified with the peptide ubiquicidin, allowing it to target the bacteria. Once there, secreted bacterial toxin released the gentamicin payload [94]. Alternatively, a thrombin sensitive peptide was successfully attached to Polyvinylalcohol (PVA) to deliver gentamicin in response to another bacterial toxin [90].

By using targeted payload release, an additional layer of control is added to an already specific system. As opposed to a passive release of a drug (i.e., diffusion controlled), which often leads to both decreased treatment efficacy and increased off target effects, modifications for targeted payload release provide a specific, active release of a drug. This ability to release drugs directly to the affected tissue impacts clinical dosing requirements, with less nanoparticle-delivered drug being necessary to provide the same efficacy as systemic delivery of the drug. Furthermore, targeted payload release also improves the safety profile of the drug. This makes a substantial difference for drugs such as doxorubicin, which is cardiotoxic [60]. Although it seems intuitive, it is worth noting that all of these modifications require surface display on the nanoparticle for efficacy, as they are entirely ineffective when buried within the molecule. Surface modifications are important moderators of the “bio–nano” interface, not only directing interaction of biomacromolecules with the bulk nanoparticle but also controling the efficiency of a nanoparticle drug-delivery system. Therefore, the importance of the selected surface modifications cannot be overstated, as they will drastically affect both safety and efficacy of a nanoparticle drug-delivery system.

## 3. Corona Development

Any discussion of the nanoparticle surface is woefully inadequate without considering the protein corona, which effectively masks the bulk nanoparticle material, stabilizes the particle, and ultimately directs the biological response to the nanoparticle (Figure 5) [52]. Despite a variety of surface modifications, nanoparticles are almost immediately (<0.5 min) and inevitably [95] coated with proteins from the biological milieu, through a process termed opsonization, to produce a protein corona [96]. Importantly, the protein corona is a dynamic entity with the composition changing based both on the biological environment of the nanoparticle, as well as the competency of the immune response. This fact may contribute to a lack of clinical translatability, which has limited the promise of nanoparticles for drug delivery. As recently reviewed by Cai et al. in 2018, binding of several of these components, including albumin and ApoJ, to the nanoparticle surface has been shown to inhibit nonspecific cellular uptake by macrophages and dendritic cells [95]. In addition to prevalent interactions with albumin, opsonins, such as fibrinogen, immunoglobulins (Ig), and complement [97], which tag the nanoparticle for phagocytosis, are key components of the protein corona and play a fundamental role in moderating the immune response. Importantly, opsonins will more readily tag positively charged particles for clearance by the Mononuclear Phagocyte System (MPS) [98]; whereas, nanoparticle coverage with negatively charged moieties has been recognized to mitigate such detrimental effects [51,99]. Hydrophobic NPs have a greater affinity for apolipoprotein, while hydrophilic particles have an affinity for fibrinogen, IgG, and albumin [51,100]. Perhaps most importantly though, nanoparticles bound by hydrophobic proteins display “danger” signals (i.e., PAMPs and DAMPs), which cause them to be more quickly cleared by the immune system [101,102,103]. Alternatively, hydrophilic particles, which decrease plasma protein absorption [51,100], appear to drastically lower the level of complement activation [52,104]. Regardless, alterations in the binding pattern and conformation of different proteins in the corona [51,52], as directed by the intrinsic properties of the underlying nanoparticle surface, can modify nanoparticle circulation time and biodistribution, ultimately, either mitigating or enhancing the function of the particle [51]. In an impactful 2014 study on the nano–bio interface, Behzadi et al. examined the effect of the protein corona on drug release from both synthesized and commercial particles, finding that the protein corona surrounding Abraxane, an FDA approved albumin-bound form of paclitaxel, effectively suppressed the burst release of drug. While the protein corona surrounding Abraxane suppresses the burst release of paclitaxel, the protein corona surrounding Doxova compromises the integrity of the particle, potentially leading to increased circulation times, but consequently compromising cellular penetration and promoting premature release of its doxorubicin payload [105]. PEGylation increases the circulation time of the nanoparticle, as well as its affinity to immune-competent proteins, [51,106,107]. This effect is directed by both the surface density and molecular weigh of PEG [51,108]. Alternatively, coating with pluronic F127 has also been demonstrated to decrease serum protein adsorption, while increasing dispersion [109]. Finally, more neutral or zwitterionic particles and those above 50 nm lead to a decreased protein corona, resulting in a reduced chance of removal from the bloodstream [110,111]. Hence, it is clear that the corona, which can be altered by several nanoparticle properties [112], including bulk material, surface curvature (i.e., geometry), size [113], surface charge/hydrophobicity, and route of administration (Figure 5), dictates the true in vivo particle response [97]. The ubiquitous protein corona effectively replaces the nanoparticles’ well-characterized “synthetic identity” with a “biological identity”. A full discussion of the protein corona’s impact is beyond the scope of this review; however, several excellent review articles were recently published [51,95]. Nonetheless, when choosing surface modifications, the protein corona must be included in considerations; otherwise, higher clearance rates may result.

### Mechanisms of Clearance

The ability for a nanoparticle to reach its intended destination is almost always a direct result of the interaction between the nanoparticle’s protein corona and the biological system, specifically the MPS and the adaptive immune system [114]. The MPS controls both macrophage and monocyte deployment and, therefore, the clearance of nanoparticles [115]. Three main methods of overcoming the MPS system have been explored: (1) passive, by increasing half-life; (2) active, by escaping uptake; or (3) using capture as an advantage. The method of particle uptake and relationship with the MPS are key factors in determining which surface modification(s) should be incorporated in the nanoparticle design. Additionally, the specifics and tendencies of the MPS can vary based on the target tissue, making target tissue and route of nanoparticle administration critical context for the design of nanoparticle drug-delivery systems.

Upon detection of a nanoparticle by a macrophage, the process of “ingestion”, via either phagocytosis or pinocytosis, begins. Phagocytosis is the primary mechanism of nanoparticle clearance as most nanoparticles are between 50 and 200 nm in size [116,117]. Macrophage phagocytosis has four broad stages: recognition, stage I, stage II, and degradation [118]. Throughout each stage, particles are ingested based on size, shape, and fluid dynamics [118], hinting that nanoparticle surface modification may play an influential role in the process. Notably, in certain cases, receptor mediation identifies antigens exposed in the protein corona, allowing specific recognition prior to phagocytosis, slowing the process [118]. Four receptor families compose the majority of macrophage cell surface receptors, including toll-like receptors, mannose receptors, scavenger receptors, and Fc receptors [116]. Nanoparticles are hypothesized to be susceptible to macrophage pattern recognition in a manner that is very similar to pathogen pattern recognition, specifically through DAMPs and PAMPs, as previously discussed [116]. Beyond simple protein pattern recognition, the mechanism of nanoparticle sorting remains enigmatic but is hypothesized to be connected to three main processing events: (1) cell-autonomous antimicrobial defense mechanisms, (2) native pathogenic or foreign material cellular process mechanisms, or (3) opsonization recognition events [116]. Once particles have been shuttled through recognition and at least half of stage I, breakdown of the particle is inevitable [118]. However, if a nanoparticle escapes before half of its circumference is engulfed, there is a slight chance that the particle can return to circulation. A few select studies have demonstrated this potential. In one particularly intriguing study, the liver was saturated by using phosphatidyl-choline: cholesterol (PC: Chol) nanoparticles [119]. PC: Chol liposomes were injected via tail-vein; 1.5 h after injection, drug-encapsulating nanoparticles were injected [119]. The results showed an increased half-life of the nanoparticle-encapsulated drug, leading to the hypothesis that PC: Chol nanoparticles saturated the liver, overwhelming liver resident macrophages and impairing their ability to uptake the drug nanoparticles [119]. Approximately 99% of nanoparticles are engulfed and eliminated by liver resident macrophages [120]. While this is an interesting hypothesis, much work remains to be done, particularly on the safety of such an approach to increase the circulation time of particles. Notwithstanding this thought-provoking application, stage I is largely a passive process governed by the innate interactions between the size and shape of the particle and the fluid dynamics surrounding the nanoparticle [118]. At the completion of stage I, stage II rapidly occurs with the involvement of active protein pathways. Finally, during degradation, the invaginated, membrane bound nanoparticle fuses with a lysosome or endosome to complete the degradation and elimination cycle.

The second possible path for nanoparticle uptake is pinocytosis, which has several subcategories of pathways that can occur, including clathrin-mediated, caveola, and clathrin/caveola [121,122,123,124]. These pathways differ from phagocytosis in the lack of a stage I, zipper-like envelopment, which is unnecessary because of the nanoparticles’ smaller size [116,121]. Pinocytosis largely deals with materials that do not fall within the typical patterns of engulfment, which occur according to size and shape, because they possess atypical surface chemistry [116]. The dependence of phagocytosis on protein pattern recognition and pinocytosis on surface chemistry describes the main obstacle for therapeutic nanoparticle development. Utilizing common surface modifications (e.g., PEGylation) in the field of therapeutic nanoparticle development may initially improve the circulating half-life of the particle; however, the complexity of the immune response and duality of clearance mechanisms may merely prime the immune system for a subsequent adverse reaction.

While additional work remains to be done to understand and control the complex nano–bio interface, the current body of literature can be dissected to reveal general trends for effective design of nanoparticle based drug-delivery systems. Certain nanoparticle characteristics have consistently demonstrated increased circulatory residency time. Size is perhaps the most influential factor when optimizing nanoparticle clearance for therapeutic efficacy. Nanoparticles 40 nm or larger take the longest time to clear (≥6 months), while nanoparticles below 15 nm clear within 24 h [125,126,127]. Closely related to the size of the particles is the shape. Experimental evidence suggests that spherical nanoparticles are less likely to accumulate in the liver or spleen when compared to rod-shaped nanoparticles [128]. Surface charge and hydrophobicity are not only major determinants of the protein corona, but are also influential in enhanced circulatory residence. Importantly, this effect is difficult to distinguish from the effect of charge and hydrophobicity on the development of the corona and the macrophage response to such [129,130]. In fact, most, if not all, of the effects of nanoparticle surface chemistry on the particle’s clearance can be directly linked to the formation and composition of the protein corona and the subsequent macrophage response.

## 4. In Vivo Nanoparticle Assessment

An accurate evaluation of the safety and efficacy of a nanoparticle drug-delivery system requires careful consideration of the appropriateness of preclinical animal models, based not only on the disease state and desired ROA, but also on the physiology of the model. Surface chemistry and functionalization play a significant role in the nanoparticle’s absorption, distribution, metabolism, and excretion (ADME), as well as potentially dictating the appropriate ROA [131]. Hence, in vivo nanoparticle performance in preclinical animal models can be condensed to be a thorough evaluation of toxicology, biodistribution, clearance kinetics, and efficacy of drug delivery [132]. A delay in clearance kinetics due to nanoparticle surface functionalization is known to increase biodistribution to target tissues [133,134], decrease toxicity [22,125,135], and increase overall efficacy, properly underscoring the importance of accurate in vivo evaluation. While tailoring the physical characteristics of the nanoparticle is crucial for safe and effective payload delivery [22,136], the preclinical animal model is arguably equally important [137,138,139]. Accurate prediction of retention, efficacy, and clearance in human patients by using animal models remains a challenge [137,138,139,140]. Just as clearance properties of nanoparticles are engineered based on the targeted disease state, route of administration, and desired release kinetics, selection of animal models must be driven by the same criteria for successful translation into human patients.

### 4.1. A Comparison of Preclinical In Vivo Models

Due to the vast array of studies, there are numerous animal models; however, it is critical to select the best possible experimental design. This section discusses the comparative anatomy and physiology of several possible animal models, concluding in a proposed preclinical development pathway (Figure 6).

#### 4.1.1. Zebrafish

According to the Guide for the Care and Use of Laboratory Animals, initial in vivo studies for novel nanoparticle formulations are tested in the lowest order vertebrate animal model, allowing for high throughput, minimal cost, and clinical relevance. Recognizing these criteria, an increasing body of evidence points to embryonic and adult zebrafish as an ideal model for initial toxicity and clearance studies [36,120,141,142,143]. The zebrafish’s translucent body allows for in vivo imaging techniques not possible in mammalian models, such as fluorescence microscopy [36,120,142,144,145] and two photon multifocal microscopy [142]. The similarity in nanoparticle clearance mechanisms is thought to result from a conserved innate and adaptive immune system between adult zebrafish and mammals [146,147].

Although a limited number of studies in zebrafish have been done, this model has demonstrated many advantages for initial testing of newly developed nanoparticle systems [141]. It is well-known that nanoparticles decorated with PEG are partially shielded from opsonization and phagocytosis, resulting in increased circulation time in mammals [22], as previously discussed. Recently this effect was discovered to be conserved in the zebrafish model [36,144,145], paving the way for the most common surface modifications to be quickly evaluated. One study by Sieber et al. supported this conclusion with the use of transgenic zebrafish embryos expressing green fluorescent protein in their vasculature and green fluorescent KAEDE protein in their macrophages [36]. The group intravenously injected 1,1′-Dioctadecyl-3,3,3′,3′-tetramethylindocarbocyanine perchlorate sterol labeled liposomes with varying PEG molecular weights and densities into the zebrafish. They were able to monitor agglomeration in the vasculature and co-localization of the particles and macrophages by confocal laser scanning microscopy [36]. To represent macrophage clearance, the particles were monitored at both 3 and 24 h post-injection and were compared to the splenic clearance in a rat model [36]. The study concluded that clearance due to macrophages in zebrafish has a high level of correlation with splenic clearance observed in rats, with increasing PEG chain length and density resulting in a lower degree of macrophage uptake [36]. A similar study by Evensen et al. demonstrated similar results in a zebrafish model, which had been seeded with fluorescently labeled human cancer cells, giving rise to the formation of tumorlike structures [145]. Interestingly, the PEGylated liposomes used in this study also passively accumulated in the tumorlike tissues, similar to murine models [145]. This model can be implemented with the addition of cancerous tumors, as well; however, further investigation using various surface modifications and tumor cell lines are needed to fully understand the clinical translatability of this strategy. Nevertheless, the collective body of literature supports the use of zebrafish as a model organism to study nanoparticle clearance and toxicity. Importantly, the use of zebrafish offers the further advantage of having well-defined tools for genetic and cellular manipulations.

#### 4.1.2. Rodents

After successfully evaluating nanoparticle toxicity in a zebrafish model, a stepwise approach to evaluate the same particles in a higher-order mammalian vertebrate, which can better replicate the disease pathology, is necessary. Rodent models, specifically mouse models, are the most well-established preclinical models for nanoparticle assessment. Despite their history, choosing the appropriate mouse strain based on its immune-system bias has drastic implications on the interpretation of safety and efficacy of data [148]. This is especially important considering that a disproportionate number of nanoparticles are being developed, particularly for immunotherapy [149], to target cancer through IV administration. A vast number of nanoparticle studies implement athymic or nude mice for cancer studies, allowing for rapid and unencumbered tumor growth upon inoculation of cells. Notably, NOD Skid Gamma mice (NSG) are gaining popularity for such studies because of their increasing susceptibility to tumor formation in Patient Derived Xenograft (PDX) models, a feature not as prevalent in a nude mouse model. This seemingly positive feature of both the nude mouse and NSG models is now under increasing scrutiny due to a physiologically compromised immune system (Table 2), which potentially affects nanoparticle clearance outcomes, an affect amplified upon repeated administration due to deficient adaptive immunity [148]. Both NSG and nude mice are T-cell deficient; however, nude mice may potentially develop mature T cells when older than six months of age. This deficiency causes a significantly decreased circulating concentration of IgG and IgA, which are both recognized to opsonize nanoparticles [150,151]. IgG is known to have a significant effect on the opsonization process, including C3 protein deposition and subsequent clearance by macrophages [152], while IgA is currently under investigation for its role in macrophage uptake [107]. Both the lack of opsonizing blood proteins and compromised adaptive immunity should be considered as possible confounding factors when interpreting nanoparticle clearance data derived from these mouse models. A study by Muller et al. comparing plasma corona formation from various animal models by using a variety of different nanoparticles highlighted this complication [140]. The group concluded that nanoparticles should be incubated in plasma from the intended animal model, to predict transferability from in vitro to in vivo studies; data derived from these studies should then be compared with analogous studies using human plasma, due to the interspecies variability of corona [140].

Although these immune-compromised mouse models may be very effective when reproducing disease states, immune-competent mice should be considered as a more reliable murine model for nanoparticle assessment. Importantly, not all immune-competent mice are equally competent when considering nanoparticle clearance [153]. In a study by Jones et al. (Figure 7), it was discovered that Th2 biased mice (BALB/c and DBA/2) cleared particle replication in non-wetting templates (PRINT) nanoparticles at a more rapid rate than Th1-biased mice (C57BL/6 and B10D2) [153]. This study also concluded with an in vitro study using human macrophages conditioned with either Th1 or Th2 cytokine mixtures, resulting in a similar increased uptake in Th2-conditioned cells [153]. While healthy humans should produce a balanced Th1 and Th2 immune response, this balance may be shifted in response to nanoparticle administration [154]. Due to this variable response, both Th1 and Th2 biased murine strains should be considered for clearance studies (Table 2). In a similar study using PRINT nanoparticles, it was concluded by Kai et al. that tumor presence increases M2-like macrophages, leading to increased clearance [155]. Notably, Th2 biased strains should be considered the most stringent when examining in vivo nanoparticle clearance.

### 4.2. Route of Administration

The desired nanoparticle route of administration (ROA), which can vary depending on disease state, surface modifications, and the material composition of the nanoparticle itself, must be considered when choosing the appropriate animal model for preclinical development. Typical ROAs used in common drug-delivery practice include inhalation, systemic, intra-articular, subdermal, and oral. The most common routes of nanoparticle administration are intravenous injection (IV) and oral intake. For nanoparticles larger than 10 nm, which mitigates renal clearance [132,156], IV and oral routes are susceptible to rapid clearance by the mononuclear phagocyte system, necessitating surface modifications for increased circulation and retention times, as previously discussed [132,157]. Nevertheless, increasingly, nanoparticles are being utilized not only to address a variety of disease states but also to access previously undruggable targets, requiring administration through ROA’s that provide unique challenges when choosing the proper animal model. Pulmonary delivery, for example, can be used for both systemic and local delivery of nanomedicine; however, unique clearance mechanisms are encountered. The inhalation of nanoparticles results in initial mucociliary clearance in the upper airway, and further clearance by lung-associated macrophages (LAM) in the lower airway [158,159,160]. Rats represent the predominant species in which inhaled nanoparticles are tested [161,162,163], although, surprisingly, rat pulmonary clearance and toxicity have not been extensively studied in comparison to humans [164]. Only recently, pulmonary clearance of 20 nm gold nanoparticles was annotated in a rat model, with clinical comparisons being made [163,165]. The group concluded that gold nanoparticles were rapidly removed by mucociliary clearance, while alveolar macrophages were responsible for long-term clearance in a rat model, similar to the expected human response [163,165]. The particles also appeared to display analogous air–blood barrier translocation when compared to humans [163].

Testing nanoparticles by using an inhalation ROA, although common, presents unique challenges. Intra-articular (IA) delivery of nanoparticles is becoming an increasingly attractive ROA to image synovial joints [166] and deliver a drug payload [167,168]. In October 2017, ZILRETTA, a triamcinolone acetonide encapsulated PLGA co-polymer matrix microspheres, (Table 1) was approved for intra-articular delivery [33]. This alternative ROA posses a unique set of challenges when choosing an animal model, including joint size, anatomy, and loading mechanics. While rodent joints are quite small—drastically increasing the technical difficulty for intraarticular delivery studies—they are commonly used [167,169,170,171]. The most common joint manipulated in rodent models is the knee, due to its large relative size and pertinence to osteoarthritis, the most prevalent joint disease [172]. Mouse and guinea pig models are also excellent for osteoarthritis (OA) studies due to the availability of knockout and strains predisposed to spontaneous OA [169,173]; however, rats are the predominant small animal model for IA nanoparticle studies—even being used for initial studies for ZILRETTA [167,170,171]. Though rat models are an excellent initial model for nanoparticle studies, showing relatively similar results in humans for cases such as ZILRETTA [171], a larger animal, with physiology and biomechanics more closely resembling human joints, should be used to fully understand clinical translatability [169].

## 5. Conclusions

Pervasive interactions with a vastly complex immunological system, including the MPS, have resulted in the rapid clearance of the majority of nanoparticles, representing the primary obstacle for the clinical translation of nanoparticle drug-delivery systems. An increasing number of studies are trying to tease apart these complex interactions by using nanoparticle surface modification. Surface functionalizations such as PEGylation [9,22,174,175], surface protein addition [4,176,177,178,179], overall charge [180,181], and pH or ion sensitivity [10,78,181,182,183,184] are being designed and synthesized to delineate the physiological target and increase the particle’s retention time within the body. Regardless of the specific modifications, characterization of these particles is of indisputable importance and should include an adequate demonstration of the quality of nanoparticle chemistry and consistency in manufacturing prior to proceeding to the clinic [131]. Furthermore, while modifications are critical for the efficacy of nanoparticle-based drug delivery, they must be carefully selected to balance the biological response to the particle and the desired physiological location of delivery. Selection must also consider the bulk nanoparticle material, disease state that could compromise efficacy or elimination of the particle, and immune system clearance. In spite of the promise of nanoparticle drug delivery and the intense development efforts representing millions of dollars in taxpayer funding, very few particles have successfully navigated the FDA approval process, casting doubt not only on the translatability of the technology but perhaps demanding more scrutiny of the experimental nuances that lead to a clinically translatable technology. This review strives to not only highlight several promising surface modifications but also to provide some general “rules of thumb” when selecting both nanoparticle surface modifications and the appropriate preclinical animal model to assess the immune response to those modifications.

## Figures and Tables

**Figure 1 ijms-20-06056-f001:**
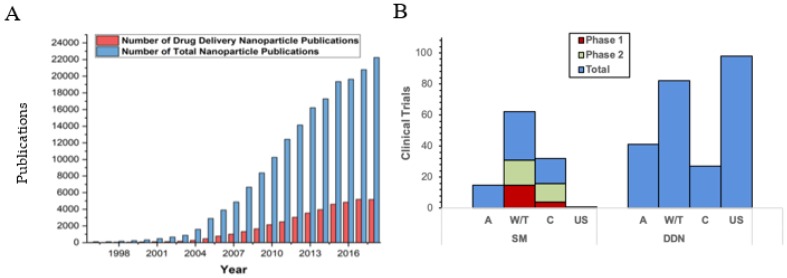
(**A**) Nanoparticle publications in PubMed as of 1 October 2019. (**B**) Nanoparticles currently in clinical trials (clinicaltrials.gov). There are currently no phase III clinical trials for drug delivery nanoparticles. Surface Modified (SM), Drug Delivery Nanoparticles (DDN), Active trails (A), Withdrawn/Terminated (W/T), Completed (C), Unknown Status (US). Note that SM is a subset of DDN.

**Figure 2 ijms-20-06056-f002:**
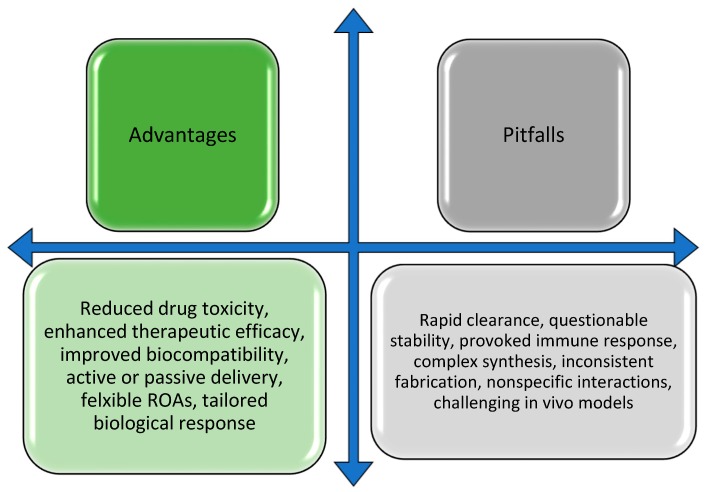
Advantages and pitfalls of nanoparticle drug-delivery systems.

**Figure 3 ijms-20-06056-f003:**
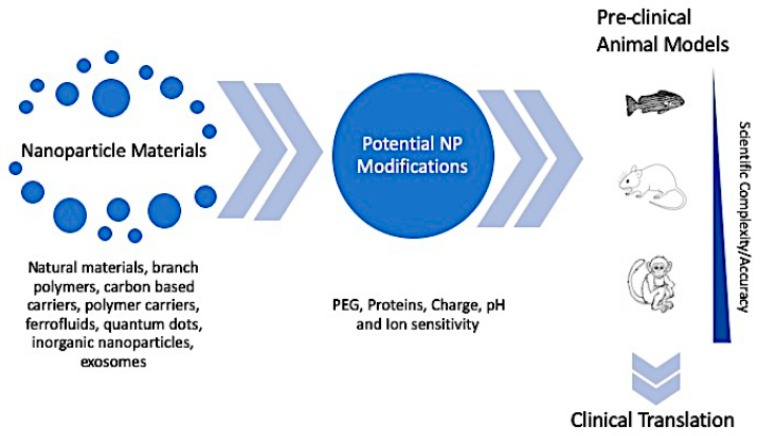
Process of nanoparticle drug delivery development.

**Figure 4 ijms-20-06056-f004:**
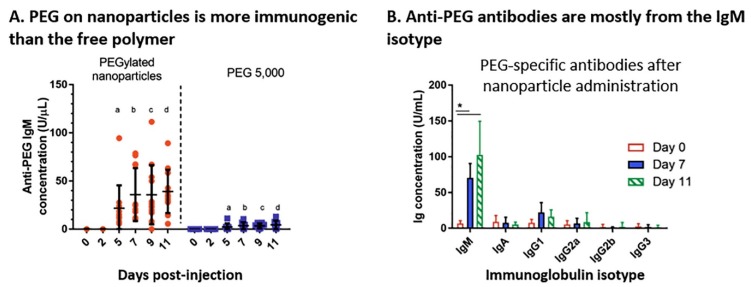
PEGylated nanoparticles show a stronger production of anti-PEG immunoglobulin M (IgM), without affecting the overall concentrations of other antibodies. (**A**) The titers of anti-PEG *IgM* increase five to seven days after an injection of PEGylated nanoparticles, whereas the injection of methoxy-terminated PEG 5000 is less immunogenic. Values represent individual animals (*n* = 9–12), a, b, c, d *p* < 0.05 between each other. (**B**) PEG-specific antibodies are mostly of the IgM isotype. Values represent means ± SD (*n* = 7–8), * *p* < 0.05. Used with permission from Journal of Controlled Release [48].

**Figure 5 ijms-20-06056-f005:**
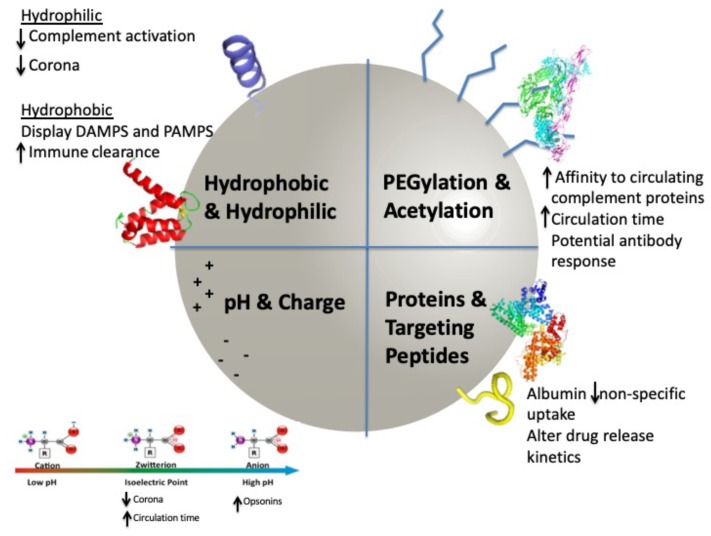
Nanoparticle surface modifications have a significant impact on a variety of aspects of the protein corona.

**Figure 6 ijms-20-06056-f006:**
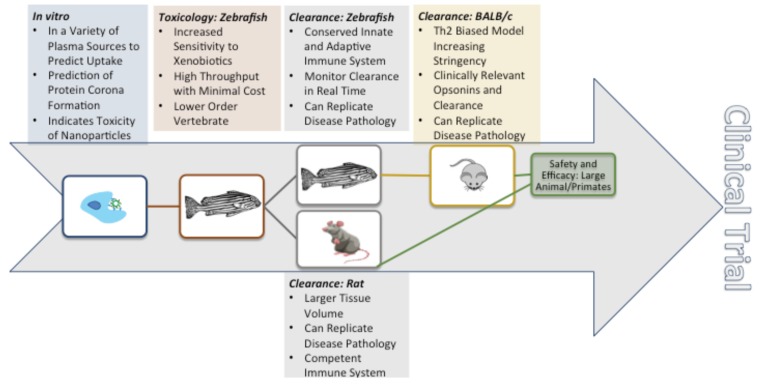
Clinical development pathway showing the advantages and order of animal models available.

**Figure 7 ijms-20-06056-f007:**
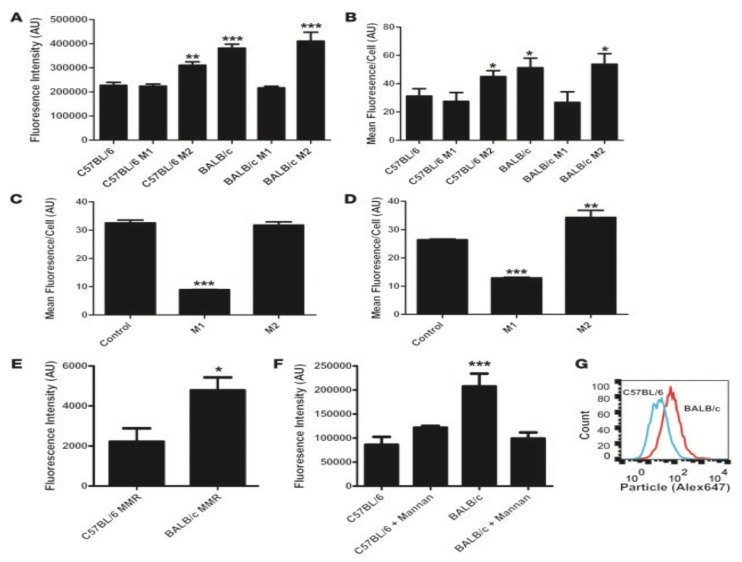
(**A**) Average integrated fluorescence per cell showed a significant (*p* < 0.0001, 1-way ANOVA with Dunnett’s post test) increase in particle uptake by BALB/c untreated, BALB/c Th2-treated, and C57BL/6 Th2-treated versus C57BL/6 untreated cells. C57BL/6 M1-treated cells showed no significant difference compared with C57BL/6 untreated cells (n = 4). (**B**) Flow cytometric analysis of uptake showed a significant (*p* < 0.05, 1-way ANOVA with Dunnett’s post test) increase in uptake by BALB/c untreated, BALB/c Th2-treated, and C57BL/6 Th2-treated versus C57BL/6 untreated cells. C57BL/6 M1-treated and BALB/c M1-treated cells showed no significant difference compared with C57BL/6 untreated cells (n = 4). (**C**) Flow cytometric analysis of uptake by human macrophages from volunteer A. M1 macrophages took up significantly fewer particles than control or M2 macrophages (n = 4) (*p* < 0.0001, 1-way ANOVA with Dunnett’s post test). (**D**) Flow cytometric analysis of uptake by human macrophages from volunteer B. M1 macrophages took up significantly fewer particles than the control macrophages (*p* < 0.0001, 1-way ANOVA with Dunnett’s post test). M2 macrophages took up significantly more particles than the control macrophages (n = 4) (*p* < 0.001, 1-way ANOVA with Dunnett’s post test). (**E**) Flow cytometric analysis of surface MMR expression on Balb/c and C57BL/6 mice. Balb/c mice showed significantly higher surface expression of MMR (n = 4) (*p* < 0.05, unpaired 2-tailed t test). (**F**) Microscopic analysis of uptake by BMMs after mannan blocking. The addition of mannan reduced uptake by Balb/c macrophages to the same levels as those of the C57BL/6 controls (n = 4) (*p* < 0.0001, 1-way ANOVA with Dunnett’s post test). (**G**) Representative flow cytometric histogram of particle uptake by BALB/c and C57BL/6 BMMs.

**Table 1 ijms-20-06056-t001:** Nanoparticle formulations that are either FDA approved or clinical trials for a targeted disease state. These nanoparticles are using compounds such as PEG, mPEG, and PGLA to increase their circulation time.

Product Name	Formulation	Stage of Development	Targeted Disease State	Source
Mircera	Methoxypolyeletheneglycol-epoetin beta	FDA Approved	Anemia, chronic renal failure	[21]
PegIntron	Peginterferonalpha-2b (mPEG-interferon alpha-2b)	FDA Approved	HIV inflammation	[24]
Macugen^®^/Pegaptanib	PEGylated anti-VEGF aptamer (vascular endothelial growth factor) aptamer	FDA Approved	Anemia with chronic renal failure	[25]
Pegasys	PEGylated IFN alpha-2a protein	FDA Approved	Hepatitis B and C	[26]
Neulasta^®^/pegfilgrastim	PEGylated GCSF protein	FDA Approved	Leukopenia by chemotherapy	[27]
Somavert^®^/pegvisomant	PEGylated HGH receptor antagonist	FDA Approved	Acromegaly	[28]
Oncaspar^®^/pegaspargase	Polymer-protein conjugate PEGylated l-asparaginase	FDA Approved	Acute lymphocytic blood clot	[29]
Krystexxa^®^/pegloticase	Polymer-protein conjugate (PEGylated porcine-likeuricase)	FDA Approved	Chronic gout	[30]
Plegridy	Polymer-protein conjugate (PEGylated IFNbeta-1a)	FDA Approved	Multiple sclerosis	[31]
ADYNOVATE	Polymer-protein conjugate (PEGylated factor VIII)	FDA Approved	Hemophilia	[32]
ZILRETTA	triamcinolone acetonide encapsulated PLGA co-polymer matrix microspheres	FDA Approved	osteoarthritis-related knee pain	[33]
N/A	^124^I-cRGDY-PEG-C dots	Clinical Trials	Cancer imaging	[34]
Caelyx/Doxil	Pegylated liposomal doxorubicin HCL	Clinical Trials	Ovarian Epithelial Cancer, Fallopian Tube Cancer, or Primary Peritoneal Cancer	[35]

**Table 2 ijms-20-06056-t002:** Comparison of key immune regulators in Nude, NSG, and wild-type C57/Bl6 mice.

Strain	Mature T Cells	Th1/Th2	Mature B Cells	Immunoglobins		Macrophages	Rate of Clearance
				IgG	IgA		
0	-	-	-	-	-	Defective	↓↓
Nude	-	-	Present/ Defective	-	-	+	↓
C57/Bl6	+	Th1 Biased	+	+	+	+	↑
BALB/c	+	Th2 Biased	+	+	+	+	↑↑

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
