# Peer review of "Overcoming Hurdles in Nanoparticle Clinical Translation: The Influence of Experimental Design and Surface Modification"

_ijms, 2019, doi:10.3390/ijms20236056_

Round 1

Reviewer 1 Report

The authors described the problems and the most common routes to come from basic nanoparticle development to clinical application. The paper is very interesting and properly written, giving adequate references throughout the whole paper. Therefore I recommend that the paper is published in the International Journal of Molecular Sciences after some minor revisions are addressed.

1 – Figures should appear after their first citation within the text.

2 – Title of table 1 and 2 should be above the table

3 – In line 129 there are 2 references with a different style; line 192 “ignores”;

4 – The authors could reproduce with the appropriate permission figures from other papers to illustrate some of the stated examples.

Author Response

Reviewer #1:

The authors described the problems and the most common routes to come from basic nanoparticle development to clinical application. The paper is very interesting and properly written, giving adequate references throughout the whole paper. Therefore I recommend that the paper is published in the International Journal of Molecular Sciences after some minor revisions are addressed.

1 – Figures should appear after their first citation within the text.

Figures and tables now appear after their first citation within the text.

2 – Title of table 1 and 2 should be above the table

Titles of tables 1 and 2 are now above the table.

3 – In line 129 there are 2 references with a different style; line 192 “ignores”;

Reference styles were corrected. “Ignores” was changed to “ignore”

4 – The authors could reproduce with the appropriate permission figures from other papers to illustrate some of the stated examples.

Two examples of primary data figures were added.

Reviewer 2 Report

Overcoming Hurdles in Nanoparticle Clinical Translation: The Influence of Experimental Design and Surface Modification

Overall the review reads fragmented, tackling big subjects in relatively little space. As the authors explain in the introduction, there has been tremendous amount of papers dedicated to nanoparticle based drug delivery. To review that, is a difficult task. I find the review would be much stronger if there was a focus on one or the other topic; surface modifications or preclinical design of experiments. Where in my opinion, the latter topic is something that would be most useful for the field. There are already many more comprehensive reviews on nanoparticle surface modifications.

I also miss the connections between the paragraphs; how do the surface modifications discussed effect preclinical development and clinical translation.

The first sections seem very negative, naming issues but not giving examples of clinical successes, which I feel gives an unfair view of the field (see also specific comments below)

Specific comments:

Figure 1: what is the link with reduced publications in drug delivery and their clinical translation? Nanoparticles have found applications in many fields; e.g. sensors, bio-imaging, energy fields, so this seems an unfair association. This is again referred to in the last sentence of that paragraph:

 “This review explores the limitations that have plagued the clinical translation of nanoparticles, specifically discussing the influence of the immune system on nanoparticle drug delivery based on the most modern delivery designs and the culminating impact upon tissue targeting as well as the translatability of biomedical nanoparticle research (Fig. 1).”

It would be more fitting if the authors show the amount of nanoparticles for drug delivery went in clinical trials compared to actual clinical translation and to the amount of publications on nanoparticles for drug delivery (amount of research and resources spent versus outcome)

Line 71: target the nanoparticle, is incorrect. The nanoparticle is not targeted. This should be revised. Also the use of However is wrong in line 75. In general, there are quite some spelling mistakes or misuse of English. I would recommend to have a native English speaker proofread the manuscript.

The authors dive straight into surface modifications, but give no general introduction on nanoparticles  themselves or highlight some of the clinical successes. This gives a false impression. This should be amended.

Furthermore, indeed surface modifications is used a lot to improve nanoparticle clearance rates, this is not true for all nanoparticles, for example protein conjugates (e,g, AbraxaneTM). This should be adjusted.

The authors should explain opsonization

Line 112, specify which hurdles you refer to

Has there been a link reported on the safety risk and increased PEG antibody concentrations? If not, this should be mentioned. Also in lines 140-141, references are missing of the safety concerns regarding PEG.

Line 136 to 157 does not refer to targeting but rather continues the discussion on surface modifications and should be moved to section 2.1

In section 2.2 it would be good if the authors mention something about the active targeting strategies that are now in clinical trials. Also nanoparticle based cell targeting is a huge field, the section seems very short compared to the efforts within it.

In section 2.3 another huge topic is reduced to very few sentences, where in general very few references are given. For example at Line 217, 218, 224 references are missing

I find the two examples given in the first paragraph for responsive release not very representable. 1 is using a nanoparticle that is not that commonly used for drug delivery but more for imaging applications, the second application is only in vitro data. The authors should include more significant studies.

Line 240 how was the release facilitated?

Example on line 264 to 268: how is this enzyme responsive, all the example explains is cell and organelle targeting. If this is true it should be in section 2.1

There is no reference for figure 4in the text and fig 4 contains spelling mistakes

Figure 5 reads confusing to me, also it now reads as if all preclinical studies are done in BALB/c mice

Author Response

Reviewer #2:

Overall the review reads fragmented, tackling big subjects in relatively little space. As the authors explain in the introduction, there has been tremendous amount of papers dedicated to nanoparticle based drug delivery. To review that, is a difficult task. I find the review would be much stronger if there was a focus on one or the other topic; surface modifications or preclinical design of experiments. Where in my opinion, the latter topic is something that would be most useful for the field. There are already many more comprehensive reviews on nanoparticle surface modifications.

While there are more comprehensive reviews of surface modifications, the authors feel that if they are to discuss preclinical design of experiments it is critical to briefly review key surface modifications to provide context. Furthermore, since nanoparticle clearance, which is a product of the immune system, can be modified based on the nature of the surface modifications, this review extends the analysis of surface modifications to not only include the immune response but also purposes a path to increase the stringency of preclinical experiments thereby improving clinical translation. Thus, although a more thorough review of each subsection may be possible and may in fact already be in the current body of literature, the value of the current review lies in the connections made between surface modification, immune system response, and preclinical design of experiments. Finally, the authors do not feel that preclinical design of experiments can be adequately discussed without context of what those experiments are designed to be testing (nanoparticles and their surface modifications). Nevertheless, the authors have made several modifications throughout the written manuscript to highlight this connection and “smooth” the writing to draw reader through the logic of the discussion. 

I also miss the connections between the paragraphs; how do the surface modifications discussed effect preclinical development and clinical translation.

The connection between the sections of the paper (“how do the surface modifications discussed effect preclinical development and clinical translation”) is that in order to (1) properly assess the newly developed surface modifications and (2) draw relevant conclusions pertaining to clearance, preclinical experimental design should consider predicted clearance pathways and the immune components that mediate the clearance to best represent the human condition and improve clinical translation.

The first sections seem very negative, naming issues but not giving examples of clinical successes, which I feel gives an unfair view of the field (see also specific comments below)

The authors agree that the beginning of the review may seem negative, but we did represent clinical successes in Table 1 as well as using multiple examples throughout the manuscript.  Additionally, we have highlighted other nanoparticle successes in diagnostics and imaging and provided an additional data figures analysis from clinicaltrials.gov (Figure 1B).

Specific comments:

Figure 1: what is the link with reduced publications in drug delivery and their clinical translation? Nanoparticles have found applications in many fields; e.g. sensors, bio-imaging, energy fields, so this seems an unfair association. This is again referred to in the last sentence of that paragraph:

 “This review explores the limitations that have plagued the clinical translation of nanoparticles, specifically discussing the influence of the immune system on nanoparticle drug delivery based on the most modern delivery designs and the culminating impact upon tissue targeting as well as the translatability of biomedical nanoparticle research (Fig. 1). It would be more fitting if the authors show the amount of nanoparticles for drug delivery went in clinical trials compared to actual clinical translation and to the amount of publications on nanoparticles for drug delivery (amount of research and resources spent versus outcome)

 This review discusses the use of nanoparticles in drug delivery so other applications would be irrelevant in this context. Due to the small number of approved nanoparticles for drug delivery using a table format would not show the numbers well, which is why the table is being utilized. We feel that this figure accurately represents the vast number of nanoparticles being published versus’ the number that are for drug delivery. When the information presented in the table is also accounted for the association between all three pieces of information show the lack of clinical translation.

Line 71: target the nanoparticle, is incorrect. The nanoparticle is not targeted. This should be revised. Also the use of However is wrong in line 75. In general, there are quite some spelling mistakes or misuse of English. I would recommend to have a native English speaker proofread the manuscript.

This was amended and now reads as “target the desired tissue”. The word “however” was also removed at this point of the manuscript. The manuscript has also been carefully revised to simplify the sentence structures and eliminate grammatical errors.

The authors dive straight into surface modifications, but give no general introduction on nanoparticles themselves or highlight some of the clinical successes. This gives a false impression. This should be amended.

 The introduction was expanded and a new figure based on clinical trials of drug delivery particles was added to provide a more fair representation of the clinical translation of drug delivery particles specifically since that is the focus of the review.  Additionally, several areas of the review do indicate that nanoparticles are being used for other biomedical purposes.  Finally, the clinical successes of nanoparticles are outlined in Table 1.

Furthermore, indeed surface modifications is used a lot to improve nanoparticle clearance rates, this is not true for all nanoparticles, for example protein conjugates (e,g, AbraxaneTM). This should be adjusted.

Protein modified nanoparticles are also examined in the review, specifically Abraxane is discussed in two sections.

 The authors should explain opsonization

 This is discussed in the protein corona section, although it was not mentioned by name. This was amended and now opsonization is now defined.

Line 112, specify which hurdles you refer to

 The hurdles, which the authors are referring to were highlighted in the introduction and figure 2. We have tried to limit redundancy in the review by not outlining the hurdles at each mention, although we have tried to make it very clear throughout the manuscript.

Has there been a link reported on the safety risk and increased PEG antibody concentrations? If not, this should be mentioned. Also in lines 140-141, references are missing of the safety concerns regarding PEG.

Safety risks have now been discussed and references have been added as requested. Additionally, a primary data figure from the literature has been added (Figure 4) to highlight the production of anti-PEG antibodies.

Line 136 to 157 does not refer to targeting but rather continues the discussion on surface modifications and should be moved to section 2.1

These lines were moved to section 2.1 as requested.

In section 2.2 it would be good if the authors mention something about the active targeting strategies that are now in clinical trials. Also nanoparticle based cell targeting is a huge field, the section seems very short compared to the efforts within it.

 A new figure (1B) has been added to clarify the nanoparticles currently in clinical trials. As cell based targeted is a large field we felt that the majority of it would be best addressed in other referenced reviews as to not needlessly lengthen this review.

In section 2.3 another huge topic is reduced to very few sentences, where in general very few references are given. For example at Line 217, 218, 224 references are missing

As noted by the reviewer, surface modifications to trigger drug payload release is another large topic that cannot be completely and thoroughly reviewed within the scope of the current review; however, the authors feel should be included and well referenced. Therefore, additional sources have been added.

I find the two examples given in the first paragraph for responsive release not very representable. 1 is using a nanoparticle that is not that commonly used for drug delivery but more for imaging applications, the second application is only in vitro data. The authors should include more significant studies.

These studies were included as representative of some of the cutting edge work being done in the field; however, an additional study has been added to the manuscript to demonstrate triggered release with a more traditional type of nanoparticle.

Line 240 how was the release facilitated?

 It was pH controlled. Additional information was added to the sentence for clarity.

Example on line 264 to 268: how is this enzyme responsive, all the example explains is cell and organelle targeting. If this is true it should be in section 2.1

 The enzyme used to break the particles has been added to the sentence; however, the authors feel that this example should remain in the current section.

There is no reference for figure 4 in the text and fig 4 contains spelling mistakes

 No spelling mistakes were found in figure 4. Figure 4 has also been moved to the introduction, in which it is referenced, and is now Figure 2.

Figure 5 reads confusing to me, also it now reads as if all preclinical studies are done in BALB/c mice

Figure 5 outlines the proposed clinical development pathway discussed in the review. The figure legend now includes the term “proposed” to clarify this matter. 

Reviewer 3 Report

In this review, the authors have discuss on some of the aspects pertaining to clinical application of nanoparticles, namely the challenges revolving around biofouling, clearance and corana formation in-vivo. While the point by the authors are important, there are some major flaws that should be rectified before acceptance, especially on the many confusing sentence structures throughout the manuscript.  Overall, the reviewer felt that this work should be publishable once these weaknesses are addressed.

The introduction section should be expanded to encompass wider range of issues, namely on the influence of size as well as shape. In recent years, there had been more and more important studies on the shape of nanoparticles and this should be taken into account. Furthermore, the reviewer felt that a dedicated section on the size and shape of nanoparticle should also be included in the manuscript in order to provide a more complete picture. It may also be necessary to touch on the different materials of nanoparticles, ranging from biodegradable polymer to metallic nanoparticles such as gold. As far as the reviewer understand, gold nanoparticles had already been in clinical use and this should be included if possible considering the nature of surface modification is different compared to conventional polymeric types. There should also be a section discussing on the chemistry by which these surface modifications are performed, ranging from classical silanization, thiol-based reactions, click-chemistry etc. Clearly, section 2.1 is too short and could be expanded Please check the referencing format for the manuscript. For instance, line 129 seems to have a different referencing style. On the other hand, there is no reference in the introduction section which is somewhat strange It is also necessary to make sure that the figures provided are of higher quality. The figures in the manuscript are rather poor in quality and resembled a "cut-and-paste" job On this note, it is also necessary to ensure that proper referencing is given in the caption of these figures if these images were cited. The authors should include some figure results from other previous work on in-vivo nanoparticle studies. So far the images provided in this manuscript are of an informative nature rather than qualitative ones. It is necessary to revamp the english in this manuscript.  The structuring of sentences in the manuscript can sometimes be rather confusing. For example in line 497-500, "Although......although". These sentences should be made clearer. It is also necessary to ensure that the grammar in the manuscript be thoroughly checked again. Line 536-538 is another example of truncation in the sentence structure which renders certain sections of the manuscript unreadable at times. Line 472 "historically... is the best established". These are some of the examples that needs to be improved throughout the entire manuscript.  On this note, the reviewer would expect a thorough amendments on the english structure in the next reading of the revised manuscript.

Author Response

Reviewer #3:

The introduction section should be expanded to encompass wider range of issues, namely on the influence of size as well as shape. In recent years, there had been more and more important studies on the shape of nanoparticles and this should be taken into account. Furthermore, the reviewer felt that a dedicated section on the size and shape of nanoparticle should also be included in the manuscript in order to provide a more complete picture.

The authors agree that size and shape of the nanoparticles are significant factors and are briefly discussed (lines 101047-1053); however, the focus of this special issue is on surface modifications and hence that is the focus of this review.

It may also be necessary to touch on the different materials of nanoparticles, ranging from biodegradable polymer to metallic nanoparticles such as gold. As far as the reviewer understand, gold nanoparticles had already been in clinical use and this should be included if possible considering the nature of surface modification is different compared to conventional polymeric types.

Several examples of different nanoparticle materials are included in the review.  Gold particles are specifically discussed.

There should also be a section discussing on the chemistry by which these surface modifications are performed, ranging from classical silanization, thiol-based reactions, click-chemistry etc. Clearly, section 2.1 is too short and could be expanded

Since section 2.1 is meant to provide context, an extensive discussion of the chemistry behind the modifications is beyond the scope of the review. The authors have also slightly expanded the section and feel that the coverage is adequate.

Please check the referencing format for the manuscript. For instance, line 129 seems to have a different referencing style. On the other hand, there is no reference in the introduction section which is somewhat strange

This was amended.

It is also necessary to make sure that the figures provided are of higher quality. The figures in the manuscript are rather poor in quality and resembled a "cut-and-paste" job On this note, it is also necessary to ensure that proper referencing is given in the caption of these figures if these images were cited.

All figures have been properly referenced or are original compositions produced for this manuscript.  The authors are uncertain which figures need to be higher quality production and have attempted to make sure they are all of sufficient quality for publication.

The authors should include some figure results from other previous work on in-vivo nanoparticle studies. So far the images provided in this manuscript are of an informative nature rather than qualitative ones.

This has been amended with two primary data figures being added to the paper.

It is necessary to revamp the english in this manuscript.  The structuring of sentences in the manuscript can sometimes be rather confusing. For example in line 497-500, "Although......although". These sentences should be made clearer. It is also necessary to ensure that the grammar in the manuscript be thoroughly checked again. Line 536-538 is another example of truncation in the sentence structure which renders certain sections of the manuscript unreadable at times. Line 472 "historically... is the best established". These are some of the examples that needs to be improved throughout the entire manuscript.  On this note, the reviewer would expect a thorough amendments on the english structure in the next reading of the revised manuscript. 

The manuscript has been carefully revised for grammatical errors and sentence structures have been extensively revised for clarity.

Round 2

Reviewer 3 Report

The manuscript had been revised to the satisfaction of the reviewer.